# Effects of Dietary *Acacia nilotica* Fruit, Zinc Oxide Nanoparticles and Their Combination on Productive Performance, Zinc Retention, and Blood Biochemistry of Rabbits

**DOI:** 10.3390/ani13203296

**Published:** 2023-10-23

**Authors:** Ahmed A. A. Abdel-Wareth, Hazem G. M. El-Sayed, Abdel-Wahab A. Abdel-Warith, Elsayed M. Younis, Hamdi A. Hassan, Afifi S. Afifi, Ghadir A. El-Chaghaby, Sayed Rashad, Shimaa A. Amer, Jayant Lohakare

**Affiliations:** 1Department of Animal and Poultry Production, Faculty of Agriculture, South Valley University, Qena 83523, Egypt; 2Cooperative Agricultural Research Center, Prairie View A&M University, Prairie View, TX 77446, USA; 3Regional Center for Food and Feed, Agricultural Research Center, Giza 12619, Egypt; 4Department of Zoology, College of Science, King Saudi University, P.O. Box 2455, Riyadh 11451, Saudi Arabia; 5Department of Nutrition and Clinical Nutrition, Faculty of Veterinary Medicine, Zagazig University, Zagazig 44511, Egypt; shimaa.amer@zu.edu.eg

**Keywords:** male rabbits, nanoparticles, phytogenic, production, sustainability

## Abstract

**Simple Summary:**

With advances in scientific research and a better understanding of rabbit physiology, emphasis has shifted to optimizing feed nutritional composition and investigating novel nutritional supplements. These supplements are intended to improve performance, immunological function, and to mitigate the negative impacts of heat stress on rabbit production. The current study’s findings reveal that rabbits fed diets containing nanoparticles of zinc oxide and *Acacia nilotica* fruit powder, either alone or in combination, improved growth performance, liver and kidney functions, and zinc retention in tissues.

**Abstract:**

This study aims to examine the effects of supplementing male rabbit diets with nanoparticles of zinc oxide (Nano-ZnO) and *Acacia nilotica* fruit powder (ANFP) on production sustainability under hot climatic conditions. Eighty Californian male rabbits aged 40 days old (average body weight 738.5 ± 11 g) were divided into four treatment groups and administered one of the following diets: control diet, Nano-ZnO (50 mg/kg), ANFP (5 g/kg), or a combination of Nano-ZnO (50 mg/kg) and ANFP (5 g/kg) for a period of 60 days. Each of the 20 rabbits used in a treatment was regarded as a replicate. The results showed that adding Nano-ZnO and ANFP individually or in combination to rabbits’ diets improved (*p* < 0.05) growth performance in comparison to control. In addition, zinc contents in serum or the testis tissues in the Nano-ZnO- and ANFP-treated rabbits were significantly greater (*p* < 0.05) than those in the control group. In addition, serum levels of creatinine, alanine aminotransferase, and aspartate aminotransferase were decreased (*p* < 0.05) by supplementation of Nano-ZnO, ANFP, or their combination. Carcass criteria did not differ among the treatments. Overall, the findings of the present study indicate that rabbits fed diets containing Nano-ZnO and ANFP, as well as their combination, showed improvements in growth performance, kidney and liver functions, as well as zinc retention in tissues under hot climatic conditions. The combination of Nano-ZnO and ANFP exhibited the best performance in the rabbits. More research on the synergistic effects of Nano-ZnO and ANFP in the sustainable production of rabbit meat is required.

## 1. Introduction

To meet the increasing need for food, as well as to have a positive impact on lowering the heat stress that affects rabbits, it is urgent to find novel natural feed additives for rabbit production. Production of rabbit meat is expanding globally [1]. Given that rabbits are particularly heat-sensitive due to their small number of sweat glands and difficulty reducing their body heat, environmental circumstances continue to be significant obstacles to producing meat [2]. Animal productivity, physiological health, and reproductive efficiency are all impacted by hot climatic conditions [3,4,5,6]. One of the necessary nutrients that can be supplemented in animal diets is zinc (Zn), which may enhance immunological function, growth, bone development, nutrient retention, and enzyme structure [7,8]. Zn is a crucial trace mineral for overall metabolism [9], serving as a cofactor for numerous metalloenzymes [10], and being critical for gut health and to improve blood biochemistry, carcass criteria, and feed efficiency [8,11,12]. It also plays a substantial role in the metabolism of protein, fat, and carbohydrates [11]. Most grains such as corn, wheat, and rice used in rabbit diets, however, are high in phytates including inositol hexaphosphates and pentaphosphates, which may impede or limit the absorption of Zn [13,14]. The amount of zinc in the rabbit diets varied from 30 to 110 mg/kg in non-supplemented diets to roughly 250 mg in diets supplemented with 100 mg ZnO or 60 mg Nano-ZnO/kg [8,15,16]. Nanotechnology provides a better form of nutritional trace mineral with various properties in comparison to the bulky commercial salts [13,16]. Nanoparticles of zinc oxide (Nano-ZnO) as small as 100 nm have been created as feed additives with unique properties and activities, such as enhancing the surface area of particles and efficient uptake by cells, in contrast to other common minerals with larger particle sizes [17,18]. Supplementation of Nano-ZnO (20–80 mg/kg) improved growth performance and meat physicochemical properties, and blood biochemistry parameters in rabbits [8]. As a result, it can improve feed bioavailability. In comparison to traditional Zn sources, nanoparticles can be utilized at lower concentrations and still produce superior outcomes.

*Acacia nilotica* tree is a member of the family Mimosaceae. *Acacia nilotica* is widely cultivated in subtropical and tropical countries [19,20]. It is a significant plant in urban areas, agricultural, and pastoral systems. Due to the presence of phytochemicals, particularly polyphenols, *Acacia nilotica* exhibits antioxidant and antibacterial properties [21,22], which may enhance the productive performance of rabbits in hot climatic conditions. Furthermore, 5 g of *Acacia nilotica* fruit powder ethanolic extract showed antioxidant and antibacterial activities in our unpublished in vitro study. There is a lack of knowledge regarding *Acacia* fruits’ prospective applications in the nutrition of rabbits. The synergistic effects of Zn and *Acacia nilotica* in reducing the heat stress on rabbits’ performance is worth investigating. Therefore, the objective of this study was to evaluate and compare the effects of either Nano-ZnO or ANFP, alone or in combination, on the growth performance, trace element retention, blood metabolites, and carcass traits of male Californian rabbits.

## 2. Materials and Methods

### 2.1. Acacia nilotica Fruit Preparation and Analysis

*Acacia nilotica* tree’s fruits were plucked from a field on a farm in Qena, Egypt as required. Physically and randomly selected samples were collected, promptly weighed, and then left to dry under the sun for 48 h. Dried materials were stored at room temperature in airtight plastic containers with tight-fitting lids after being pulverized to pass through a 1 mm screen with a centrifugal mill (ZM1, Retsch, Haan, Germany) until use. The *Acacia nilotica* fruits were extracted by hydrodistillation from two batches of dried fruits in a Clevenger-type apparatus for three hours at the Regional Center for Food and Feed, Agricultural Research Center, Egypt. A mass spectrometer detector (ISQ Single Quadrupole Mass Spectrometer, THERMO Scientific Corp., Waltham, MA, USA) and gas chromatography (TRACE GC Ultra Gas Chromatographs, Thermo Fisher Scientific Corp., Waltham, MA, USA) were used for the analyses of the *Acacia nilotica* extracts. The GC-MS system included a TG-5MS column (30 m × 0.25 mm i.d., 0.25 m film thickness, manufactured by THERMO Scientific Corp. in Waltham, MA, USA). Using the following temperature program, analyses were performed using helium as the carrier gas at a flow rate of 1.0 mL/min and a split ratio of 1:10: 240 °C was reached after rising from 60 °C for one minute at a pace of 3.0 °C per minute. Both the injector and detector were kept at 240 °C. In the two tests, 0.2 µL of the mixes were injected as diluted samples (1:10 hexane, *v*/*v*). Electron ionization (EI) at 70 eV with a spectral range of *m*/*z* 40–450 was used to produce mass spectra. Utilizing analytical mass spectra (from real chemicals, the Wiley spectral library collection, and the NSIT library), majority of the compounds were identified. The average of two replicates per extract was used to express the relative quantities of active chemicals in *Acacia nilotica* fruits (Table 1).

### 2.2. Acacia Fruits Antioxidant, Total Phenols, and Total Flavonoids

The total flavonoids (TF), total phenols (TP), and total antioxidant capacity (TAC) of the ANFP were analyzed according to El-Chaghaby et al. [21]. The TF content of the extracts was determined by the aluminum chloride test using quercetin as standard and the results were calculated as mg quercetin equivalent/Kg of extract (mg/Kg). The TP content of the extracts was determined using the Folin–Ciocaleau method. The results were calculated using the regression equation of the calibration curve and represented as mg of gallic acid equivalents per kilogram of the extract (mg/Kg). The phosphomolybdenum determined the TAC of the extracts. Ascorbic acid was used as a reference antioxidant in the calculations, and the results were represented as mg of ascorbic acid equivalent per 100 g of extract (mg AAE/100 g).

### 2.3. Acacia Fruits Antioxidant, Total Phenols, and Total Flavonoids Contents

The Acacia fruits extract showed the highest levels of TAC, TP, and TF when ethanol was used as the extraction solvent (Table 2). The amount of TP in the Acacia fruit extract could be a significant predictor of its antioxidant capacity. The primary factors in the overall antioxidant potential of ANFP are the phytochemical components, particularly polyphenols. In the present investigation, pyrogallic acid (50.36%), coniferyl (10.55%), benzaldehyde (9.64%), and isovaleric acid (6.68%) were the primary components of ANFP (Table 3).

### 2.4. Preparation of Nano-ZnO

In this investigation, 800 mg/kg Zn of nano-ZnO (Cas no. 1314-13-2, Sigma-Aldrich, Steinheim, Germany) was used. The particle hydrodynamic diameter measured using dynamic light scattering was less than 100 nm. For aqueous systems, the pH was 7.5 ± 1.5 and the density was 1.7 ± 0.1 g/mL at 25 °C. Scan and transmission electron microscopes (SEM and TEM) were used to analyze the particle’s structural morphology (Figure 1).

### 2.5. Experimental Animals, Design, and Management

Rabbits were kept on a research farm at the Animal and Poultry Production Department, Agriculture Faculty, South Valley University, Qena, Egypt. The experimental protocol (SVUAGR-3-2023), ensured that the rabbits were treated throughout the trial in accordance with the guidelines for the care of experimental animals, and was approved by the committee of ethics at South Valley University. This study was carried out in compliance with the ARRIVE guidelines. Eighty Californian male rabbits (average body weight 738.5 ± 11) aged 40 days old were divided into four treatment groups and administered one of the following diets: control diet, control diet with Nano-ZnO (50 mg/kg), ANFP (5 g/kg), or a combination of Nano-ZnO (50 mg/kg) and ANFP (5 g/kg). The Nano-ZnO and ANFP were combined with a mineral premix, homogenized, and combined with feed components before being given to the rabbits as pelleted total mixed meals. Twenty replicate cages were used in each treatment. A manual feeder and an automatic drinker were included in the galvanized wire net cages, which were 44 cm in width, 50 cm in length, and 35 cm in height and were used to raise individual rabbits. During the whole 60-day trial period, rabbits were housed in similar management, sanitary, and environmental conditions. Throughout the experiment, *ad libitum* pelleted meals were provided, and automatic nipple drinkers provided access to fresh water. Table 3 provides the components of the basal diet and laboratory chemical analysis of the basal diet. The rabbits were raised in an open house system (a naturally ventilated space with windows and ceiling fans) with an average temperature of 34.5–36.5 °C, a relative humidity of 50–55%, and a temperature–humidity index (THI) of 31.38–33.42 for the entire experimental period with a 16 h light and 8 h dark regime.

The temperature–humidity index (THI) was determined using the following formula:THI = db °C − [0.31 − 0.31RH/100] (db °C − 14.4) 
where db °C stands for dry bulb temperature in Celsius and RH for relative humidity in percentage.

### 2.6. Growth Performance

At 40, 70, and 100 days of age, the initial body weight (BW), final BW, and feed intake of rabbits were all measured. Calculations of the daily BW increase and feed intake between 40 and 70, 70 and 100, and 40 and 100 days of age were performed. The feed conversion ratio (FCR) was estimated by subtracting the average daily body weight growth from the daily feed consumption. Every case of mortality and every instance of diarrhea was recorded as it happened.

### 2.7. Serum, Diet, and Tissue Zn Concentrations

Samples of the testicles, liver, and kidneys were removed from dead rabbits (*n* = 20), and they were immediately placed in freezers at –20 °C for Zn content analysis. With the use of an atomic absorption spectrophotometer (Perkin Elmer Analyst 800 model, Shelton, CT, USA), the Zn concentrations in the rabbits’ control diet, serum, liver, kidney, and testis were determined.

### 2.8. Blood Biochemical Assay

After fasting for 12 h, blood samples were obtained from each rabbit in each experimental group at the end of the study, at 100 days of age. Before centrifuging for 15 min at 3000 RPM, blood samples in tubes were kept for an hour at room temperature to allow the blood to coagulate. The sterile, dry, labeled stopper vials containing the clear supernatant serum were utilized to conduct clinical biochemical testing. Colorimetric diagnostic kits from spectrum-bioscience (Egyptian Company for Biotechnology, Cairo, Egypt) were used to measure serum creatinine, urea, aspartate aminotransferase (AST), and alanine aminotransferase (ALT).

### 2.9. Carcass Measurements

After fasting for 12 h, 20 rabbits per treatments were slaughtered (n = 20) on termination of the experimental period. The animals were weighed at slaughter to ascertain their live BW and killed humanely in accordance with halal slaughtering procedures. The carotid artery, jugular vein, trachea, and esophagus were all severed. The skin, genitals, urinary bladder, gastrointestinal tract, and distal section of the legs were taken from the slaughtered rabbits after they were bled. After emptying, the head and entire gastrointestinal system were weighed and expressed as g/kg slaughter weight. Furthermore, the weight of the entire gastrointestinal system was determined. Testes, liver, spleen, heart, lungs, kidneys, and perirenal and scapular fat were all weighed and represented as g/kg of carcass. For the dressing percentage, the carcass, head, liver, kidneys, and heart were all considered. Dressing was expressed as g/kg and proportionate to the live weight at slaughter.

### 2.10. Statistical Analysis

Using SAS 9.2′s general linear model (GLM) approach and a completely randomized design, the statistical analysis was carried out. All analyses used the pens as the experimental unit. All data were evaluated for normal distribution (W > 0.05) using the Shapiro–Wilks test. Then, Duncan multiple range tests were employed to compare means after one-way ANOVA was carried out using the SAS 9.2 [23] statistical program. The mean and standard error of the mean (SEM) were used to express values. *p* < 0.05 was used to declare significance.

## 3. Results

### 3.1. Growth Performance

The results showed that, during the entire study period, adding Nano-ZnO and ANFP individually or in combination to Californian male rabbit diets increased (*p* < 0.01) the body weight of rabbits at 70 and 100 days of age. Likewise, body weight gain was increased by supplementation of Nano-ZnO and ANFP individually or in combination, in comparison to control diet throughout the experiment phases (40–70 days, 70–100 days, and 40–100 days). In addition, feed conversion ratio was improved (*p* < 0.01) by supplementation of Nano-ZnO and ANFP individually or in combination in rabbits’ diet during 40–70, 70–100, and 40–100 days of age. When Nano-ZnO and ANFP were added to the diet of Californian males during the 70–100 day and 40–100-day periods, respectively, feed intake improved (*p* < 0.05) but did not differ between treatments during the first period at 40–70 days of age. Overall, the combination of Nano-ZnO and ANFP was found to provide the best growth performance, including body gain and feed conversion ratio (Table 4).

### 3.2. Serum Biochemical Assays

Serum Zn concentrations in the Nano-ZnO- and ANFP-treated rabbits were greater (*p* < 0.05) than those in the control group (Figure 2). Rabbits consuming diets supplemented with Nano-ZnO and ANFP individually or in combination exhibited higher (*p* < 0.001) Zn contents in the testis than those consuming the control diet (Figure 2). However, there were no significant effects of Nano-ZnO and ANFP on concentrations of Zn in liver and kidney tissue among treatments (Figure 3). In addition, serum levels of alanine aminotransferase (ALT) and aspartate aminotransferase (AST) were decreased (*p* < 0.05) by supplementation of Nano-ZnO, ANFP, or their combination compared to the non-supplemented control group (Figure 4). Likewise, serum levels of creatinine and urea were decreased (*p* < 0.05) by supplementation of Nano-ZnO, ANFP, or their combination compared to the non-supplemented control group (Figure 5).

### 3.3. Carcass Criteria

The effects on carcass criteria of rabbits due to Nano-ZnO and ANFP individually or in combination are presented in Table 5. Diets supplemented with Nano-ZnO and ANFP individually or in combination increased (*p* < 0.05) the dress-to-live body weight ratio compared to the control group under hot climatic conditions. The testes-to-BW ratio increased with the Nano-ZnO and ANFP individually or in combination compared to the control group. There were no observed differences in head, liver, heart, spleen, and kidneys to live BW ratio due to Nano-ZnO and ANFP individually or in combination compared to control.

## 4. Discussion

The literature is lacking in information regarding the potential usage of Nano-ZnO and ANFP alone or in combination in rabbit feeding and the impact on the sustainability of production of Californian male rabbits under hot climatic conditions. Therefore, comparisons were made with earlier works, as well as with other studies that investigated ZnO or *Acacia nilotica*. The antioxidant and antibacterial properties of *Acacia nilotica* are mostly attributed to its phytochemical composition, particularly polyphenols [21,22], which would sustain the rabbit production in hot climates. In the current study, Acacia fruit extract showed the highest levels of TAC, TP, and TF, which could alleviate the heat stress on animals. The amount of total phenols in *Acacia nilotica* fruits may be a significant predictor of its antioxidant capacity [17]. The phytochemical components of ANFP (Table 1), are the main cause of its overall TAC, TP, and TF.

In this study, adding Nano-ZnO and ANFP individually or in combination to Californian male diets improved the body weight, body weight gain, and feed conversion ratio of rabbits under hot climatic conditions. This improvement was partly related to a tendency to increase feed intake. In addition, Nano-ZnO and ANFP combination showed the best growth performance, which may be mostly attributable to the synergistic effects of Zn and ANFP in reducing the heat stress on rabbits. The primary phytochemical components of ANFP, particularly its polyphenol concentrations, are responsible for its capacity to enhance growth performance, stimulate appetite, and increase feed intake [17,18]. In the current study, the main components of ANFP were pyrogallic acid (50.36%), coniferyl (10.55%), benzaldehyde (9.64%), and isovaleric acid (6.68%), which could be the explanation for rabbits’ improved performance. Since there is currently no publication that discusses ANFP and its effects on growing rabbits, we compared the findings with earlier research on medicinal plants. Phytogenics in animal feed have been shown to increase appetite and feed intake, which in turn enhances growth performance [20]. Under high-temperature conditions, the growth performance of growing rabbits was enhanced due to *Acacia nilotica* bark extract feeding [21]. Additionally, adding 100 mg of ZnO to the diet could potentially assist heat-stressed growing rabbits in achieving enhanced growth performance [16]. Supplementation of Nano-ZnO at 20 mg/kg up to 80 mg/kg improved body weight, body weight gain, and feed conversion ratio of White New Zealand rabbits [8]. Similarly, Hassan et al. [18] found that when rabbits took different doses of Nano-ZnO at 30 and 60 mg/kg in comparison to the control group, growth performance and feed intake significantly improved. The level of supplementation, the sources, methods, strains, age of the rabbits, and duration of the experiments could all have an impact on the results of previous studies.

The Zn concentration in the blood and testes was higher in the rabbits’ consuming diets supplemented with Nano-ZnO and ANFP separately or in combination, but not different in terms of liver and kidney tissues. Zn has an impact on the motility of spermatozoa and is crucial for the development of the testes, spermatogenesis, and capacitation [24]. Dietary supplementation with 50 mg/kg Nano-ZnO enhanced the serum and testis zinc levels, showing increased zinc absorption. However, the absence of significant Zn contents in the liver and kidneys is mostly attributable to zinc bioavailability, and retention in these tissues. However, the literature is lacking in terms of how the combination of Nano-ZnO and ANFP affects the Zn levels in rabbit testes or blood. An earlier study showing Nano-ZnO produced by using the plant *Acacia nilotica* exhibited excellent antimicrobial activity against resistant bacteria, were harmless for animal cells, and were long-lastingly stable [25]. This might be the reason of the synergistic effects of Nano-ZnO and ANFP on increasing Zn contents in rabbit testes or blood in our study.

In the present investigation, Nano-ZnO- and ANFP-treated rabbits had decreased ALT, AST, and creatinine levels in their serum compared to control rabbits. The fact that Zn is a necessary component of roughly 200 Zn metalloenzymes and is present in a variety of enzymes that support the structural integrity of proteins may be the cause of these improvements in male rabbits’ blood biochemistry, including their liver and kidney functions [26,27]. For male sexual function, zinc is the most important trace mineral. It is necessary for the metabolism of testosterone, the health of sperm, the growth of the testicles, and the control of excessive estrogen in the tissue of the male reproductive system [28]. Additionally, because Zn regulates enzyme secretion, it may be responsible for the favorable outcomes in the liver enzymes, urea, and creatinine. Additionally, Abdel-Wareth et al. [29] reported that when Nano-ZnO at 100 mg/kg was added to the diet of rabbits, the serum concentrations of ALT and AST were much lower than in the control group. *Acacia nilotica* has shown significant antioxidant activity due to its high concentration of polyphenolic components, such as catechins, which are known to have antioxidative and anti-inflammatory effects [30] and thus improve the animal’s ability to mitigate the effect of hot climatic conditions. An acetaminophen-induced liver injury was treated with *Acacia nilotica*, according to Kannan et al. [31], and the liver enzyme levels returned to normal ranges. Consequently, it can be concluded that *Acacia nilotica* has liver and kidneys protective properties.

Furthermore, our results showed that diets supplemented with Nano-ZnO and ANFP individually or in combination increased the dressing and testes percentages without any difference in liver, kidney, and heart percentages compared to the control group under hot conditions. The control group’s live body weight (n = 20) was slightly higher before slaughter following a 12 h fasting interval (Table 5) than the final body weight at the end of the growth performance period (Table 4), but BW in the treated groups decreased after fasting, as expected. This could be due to the most important factors influencing preslaughter live weight loss are the gut clearance, water content, and physiological response of animals to fasting, which occurs during feed deprivation. Al-Sagheer et al. [32], who found a significant rise in dressing percentage in rabbits fed dietary supplements containing Nano-ZnO at 100 mg/kg, with no differences in the ratios of the liver, kidneys, and heart to body weight when compared to the control group. Male rabbits’ liver, kidney, and heart functions were unaffected by diets containing ZnO at a dosage of 100 mg/kg [33]. In comparison to the control group, dressing, liver, and kidney percentages in rabbits fed ZnO at doses of 50, 100, 200, and 400 mg/kg diet did not differ significantly [34], in line with our results.

Under high temperature conditions, adding Acacia bark extract to rabbit rations significantly boosted dressing percentages without any significant effects on internal organs [21]. Overall, the percentages of dressing and organs were comparable to those mentioned in other reports [3,35]. It appears that the absence of quantitative data on Acacia fruits impedes progress in discussing the effects on carcass characteristics in detail, although such studies should be carried out in the future.

## 5. Conclusions

The present study indicates that rabbits fed diets containing Nano-ZnO and ANFP, either alone or in combination, showed improvements in growth performance, liver and kidney functions, as well as Zn retention in tissues. The combination of Nano-ZnO and ANFP can be applied as feed additives in rabbit male diets, as it exhibited the overall best performance in the present study. Future systematic studies on the beneficial synergistic effects of Nano-ZnO and ANFP in the production of male and female rabbit meat are needed.

## Figures and Tables

**Figure 1 animals-13-03296-f001:**
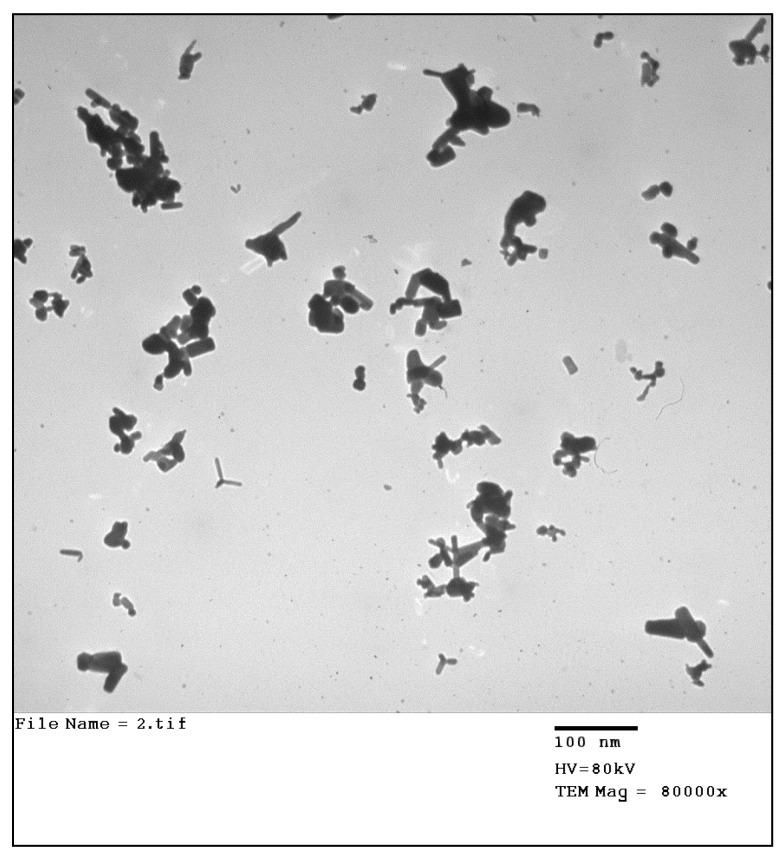
Transmission electron micrographs (TEM) of Nano-ZnO.

**Figure 2 animals-13-03296-f002:**
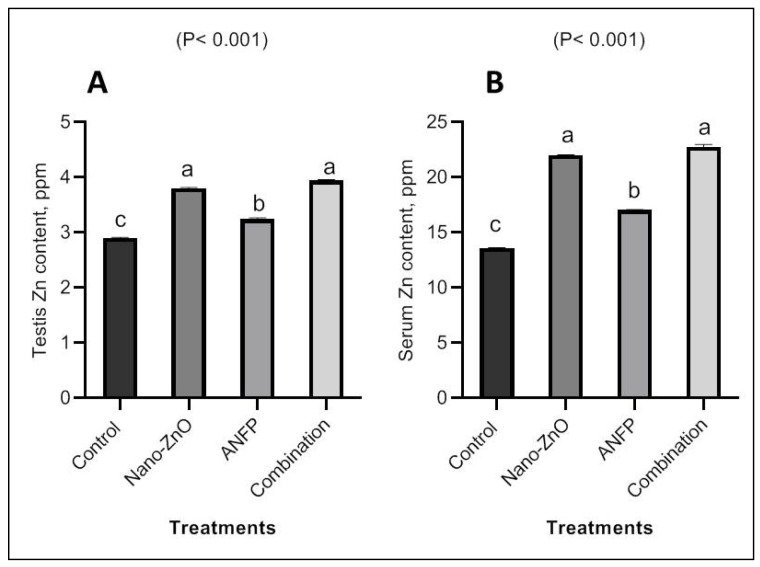
Impact of Nano-ZnO and ANFP individually or in combination on zinc content in testis (**A**) and serum (**B**) of rabbits at 100 days of age. ^a–c^ The bars in figures with different superscripts are different (*p* ˂ 0.05). ANFP: *Acacia nilotica* fruit powder. SEM: standard error of means.

**Figure 3 animals-13-03296-f003:**
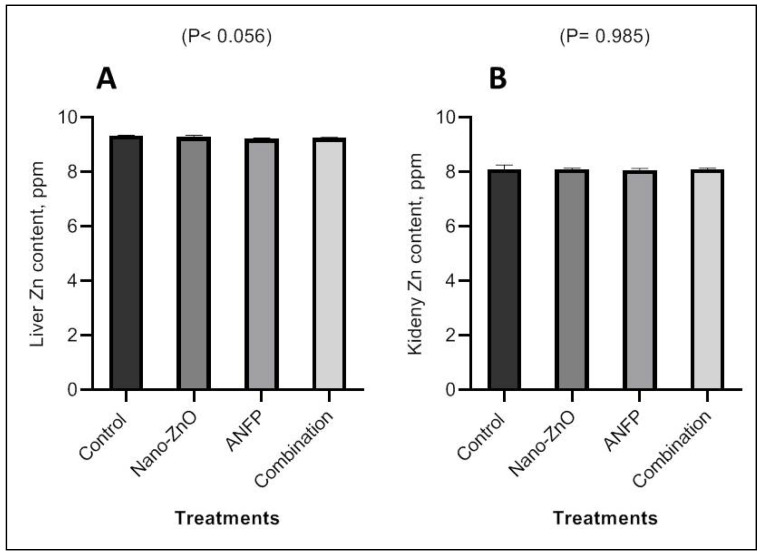
Impact of Nano-ZnO and ANFP individually or in combination on Zn content in liver (**A**) and kidneys (**B**) of rabbits at 100 days of age. The bars in figures with no superscripts are not different (*p* ≥ 0.05). ANFP: *Acacia nilotica* fruit powder. SEM: standard error of means.

**Figure 4 animals-13-03296-f004:**
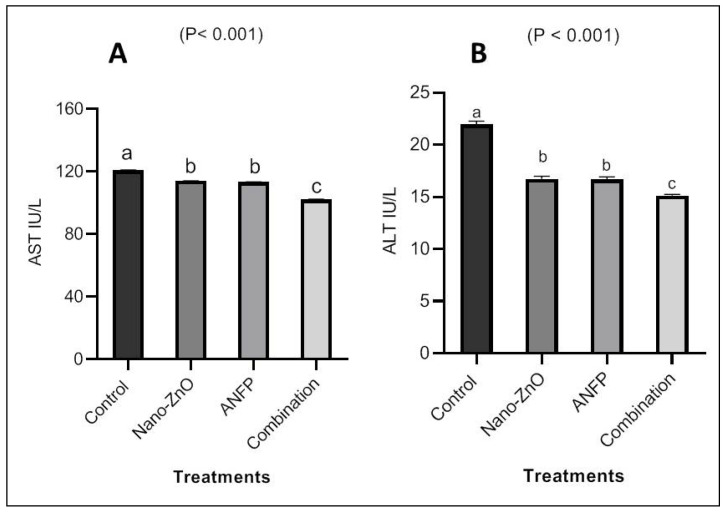
Impact of Nano-ZnO and ANFP individually or in combination on serum AST (**A**) and ALT (**B**) of rabbits at 100 days of age. ^a–c^ The bars in figures with different superscripts are different (*p* ˂ 0.05). ANFP: *Acacia nilotica* fruit powder. SEM: standard error of means.

**Figure 5 animals-13-03296-f005:**
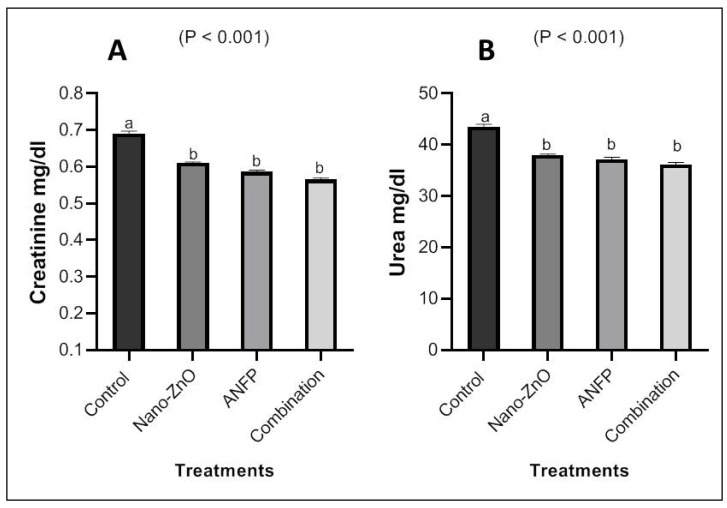
Impact of Nano-ZnO and ANFP individually or in combination on serum creatinine (**A**) and urea (**B**) of rabbits at 100 days of age. ^a–b^ The bars in figures with different superscripts are different (*p* ˂ 0.05). ANFP: *Acacia nilotica* fruit powder. SEM: standard error of means.

**Table 1 animals-13-03296-t001:** Relative composition (g/kg) of *Acacia nilotica* bioactive lipid compounds in fruits extract.

No	RT (min)	Name	Area Sum%
1.	5.63	Isovaleric acid	6.68
2.	5.48	Benzaldehyde	9.64
3.	6.21	6,7-dihydroxycoumarin	0.72
4.	7.61	4′-hydroxychalcone	2.94
5.	8.91	Coniferyl alcohol	10.55
6.	9.27	Resorcinol monoacetate	4.9
7.	9.77	3′,4′,5′,5,6,7-hexamethoxyflavone	0.84
8.	10.00	Chalcone	0.51
9.	10.35	Phytol	0.87
10.	11.44	Pyrogallic acid	50.39
11.	16.65	Palmitic acid	3.78
12.	18.11	1,2-dioleoyl-sn-glycerol	2.97
13.	18.31	Arachidic acid	0.75
14.	19.14	ω-3 arachidonic acid	1.95
15.	19.89	Glycerol 1-myristate	0.93
16.	20.96	Garcinone D	0.85
17.	21.90	Oleic acid, 3-(octadecyloxy)propyl ester	0.73

**Table 2 animals-13-03296-t002:** Antioxidant capacity, total phenols, and total flavonoids activity of *Acacia* fruit extracts.

Properties	Ethanol Extract
Total antioxidant capacity (mg AAE/100 g)	9650 ± 100
Total phenols (mg GAE/100 g)	14,127 ± 64
Total flavonoids (mg QE/Kg)	507 ± 23

**Table 3 animals-13-03296-t003:** Ingredient and chemical composition of the rabbit diet.

Ingredients	g/kg
Corn	310
Bran of wheat	200
Soybean meal (440 g/kg CP)	190
Straw of wheat	120
Hay of lucerne	50
Bran of rice	50
Straw of linseed	28
Meal of sunflower	25
Limestone	20
Sodium chloride	3
Vitamin–mineral premix ^1^	3
Dl-methionine	1
Chemical composition analyzed (g/kg, as fed)	
Dry matter	934
Digestible energy (DE, MJ/kg DM)	9.5
Crude protein	179
Ether extract	39.4
aNDFom	392
ADFom	232
ADL	69.5
Ash	93.2
Calcium	15.5
Phosphorus	7.63
Zinc	0.09

^1^ Source: Ibex International Co., Ltd., Cairo, Egypt. Vitamin A 10,000 IU, vitamin D3 900 IU, vitamin E 50.0 mg, vitamin K 2.0 mg, vitamin B1 2.0 mg, folic acid 5.0 mg, pantothenic acid 20.0 mg, vitamin B6 2.0 mg, choline 1200 mg, vitamin B12 0.01 mg, niacin 50 mg, biotin 0.2 mg, Cu 0.1 mg, Fe 75.0 mg, Mn 8.5 mg, and ZnO 20 mg per kilogram of diet. aNDFom: neutral detergent fiber tested without sodium sulfite and with a stable heat amylase, expressed only of residual ash; ADL: acid detergent lignin; ADFom: acid detergent fiber expressed exclusively of residual ash. DE = gross energy in feed–gross energy in feces.

**Table 4 animals-13-03296-t004:** The growth performance of rabbits supplemented with Nano-ZnO and ANFP individually or in combination in the diets.

Items	Treatments				SEM	*p*-Value
	Control	Nano-ZnO	ANFP	Combination		
Body weight, g						
40 days	735	737	740	742	11.23	0.264
70 days	1626 ^c^	1784 ^b^	1775 ^b^	1881 ^a^	23.18	<0.001
100 days	2508 ^c^	2782 ^b^	2750 ^b^	2876 ^a^	23.82	<0.001
Body weight gain, g						
40 to 100 days	891 ^c^	1047 ^b^	1035 ^b^	1139 ^a^	15.36	0.003
70 to 100 days	882 ^b^	998 ^a^	975 ^a^	995 ^a^	16.12	0.025
40 to 100 days	1773 ^c^	2045 ^b^	2010 ^b^	2134 ^a^	22.42	0.001
Feed intake, g						
40 to 100 days	2787	2786	2754	2790	7.19	0.250
70 to 100 days	3058 ^c^	3172 ^b^	3133 ^b^	3256 ^a^	25.17	0.001
40 to 100 days	5845 ^c^	6058 ^b^	5987 ^b^	6146 ^a^	26.15	0.001
Feed conversion ratio						
40 to 100 days	3.128 ^a^	2.661 ^b^	2.661 ^b^	2.450 ^c^	0.070	0.001
70 to 100 days	3.467 ^a^	3.178 ^b^	3.213 ^b^	3.272 ^b^	0.042	0.023
40 to 100 days	3.297 ^a^	2.962 ^b^	2.979 ^b^	2.880 ^b^	0.036	0.004

^a–c^ Means (*n* = 20) with different superscripts in a row are significantly different (*p* < 0.05). ANFP: *Acacia nilotica* fruit powder. SEM: standard error of means.

**Table 5 animals-13-03296-t005:** The carcass criteria of rabbits supplemented with Nano-ZnO and ANFP individually or in combination in the diets.

Items	Treatments				SEM	*p*-Value
	Control	Nano-ZnO	ANFP	Combination		
Carcass characteristics percentage						
Live body weight, g	2577	2656	2677	2705	52.08	0.080
Dressing %	55.97 ^c^	60.34 ^b^	61.29 ^b^	66.68 ^a^	0.955	0.005
Liver %	3.210	3.139	3.361	3.177	0.085	0.831
Heart %	0.284	0.324	0.313	0.323	0.009	0.402
Head %	4.772	5.127	5.002	4.800	0.068	0.199
Spleen %	0.054	0.057	0.054	0.058	0.005	0.376
Testis %	0.174 ^b^	0.227 ^a^	0.224 ^a^	0.229 ^a^	0.011	0.021
Kidneys %	0.775	0.854	0.722	0.722	0.029	0.333

^a–c^ Means (*n* = 20) with different superscripts in a row are significantly different (*p* < 0.05). ANFP: *Acacia nilotica* fruit powder. SEM: standard error of means.

## Data Availability

The datasets used and/or analyzed during the current study can be made available from the corresponding author on reasonable request (A.A.A.A-W.).

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
