# Peer review of "Effects of Dietary Acacia nilotica Fruit, Zinc Oxide Nanoparticles and Their Combination on Productive Performance, Zinc Retention, and Blood Biochemistry of Rabbits"

_animals, 2023, doi:10.3390/ani13203296_

Round 1

Reviewer 1 Report

The work focuses on an interesting aspect that may be of interest to rabbits housed in conditions of heat stress. The document is generally well-written. However, there are some aspects mentioned below that must be reviewed and corrected, since today they would prevent its acceptance for publication. Please, consider all the recommendations that I describe below.

In the study has not been evaluated the health status of the animals, only some traits related to the liver and kidney function. Please, delete all the mentions related to health status in the paper.

As the basal diet provides Zn throughout the raw materials and the ZnO of the premix, it justifies why did you included ZnO in the premix, the current recommendations of Zn for growing rabbits, why you have included Zn over this current recommendations…

In the introduction was mentioned that potential synergic effect with Zn sources, but it must be explained why alone acacia inclusion increased the Zn level in the animals.

Section 3.1. of the results is not a result section, I recommend including it at the material and methods section.

Were animals housed individually? The size of the cages seems to be too big for individual housing.

Please, indicate if the chemical composition was determined in your lab or provided by the feed manufacturer. Was DE value determined in a digestibility trial? Describe it.

However, Zn level of the basal diet it is very important as the paper is focused in the Zn level and source. Please, describe how you have determined the Zn in the basal diet, as well as the Zn level of the four dietary treatments.

I was surprised by the Zn value of the basal diet (90 g/kg), I believe that it is a mistake as the current recommendations for growing rabbits are 35 mg/kg. Please, review this value and justify why the basal diet includes ZnO in the premix. Define that your paper is addressed to evaluate the inclusion of nano-ZnO providing Zn over the current recommendations.

Line 181 (n=20). Does it mean that you have slaughtered all the animals of each treatment??

Justify in the introduction the need to control the effect of the treatments on both blood biochemical and carcass traits. 

Statistical analysis of performance traits (BW, ADG, DFI and FCR) must be done using a repeated measures procedure (mixed model), in order case, you are providing the effect of the animal within the treatment to treatment, overestimating the effect of the treatment.

All the tables and figures must be self-explanatory, please review the use of abbreviations in these elements.

Avoid the use of the term “demonstrate” by “show”.

In the discussion section there is a lack of comparison with the current recommendations of Zn and other previous studies where Zn level in rabbits has been evaluated. It is important due to the different level used in this study.

The English language is not so bad, but it could be reviewed by a native.

Author Response

Thank you very much for taking the time to review this manuscript. Please find the detailed responses below and the corresponding revisions/corrections highlighted in track changes in the resubmitted files. We addressed all suggestions and comments.

Response to Reviewer 1 Comments

Comments 1: The work focuses on an interesting aspect that may be of interest to rabbits housed in conditions of heat stress. The document is generally well-written. However, there are some aspects mentioned below that must be reviewed and corrected, since today they would prevent its acceptance for publication. Please, consider all the recommendations that I describe below.

Response 1: Thank you very much for taking the time to review this manuscript. Please find the detailed responses below and the corresponding revisions/corrections highlighted in track changes in the resubmitted files. We addressed all suggestions and comments.

Comments 2: In the study has not been evaluated the health status of the animals, only some traits related to the liver and kidney function. Please, delete all the mentions related to health status in the paper.

Response 2: Thank you for pointing this out. The mention of the “health status” is now deleted throughout the manuscript as suggested. Please see Line 41, and throughout the manuscript, it is deleted.

Comments 3: As the basal diet provides Zn throughout the raw materials and the ZnO of the premix, it justifies why did you included ZnO in the premix, the current recommendations of Zn for growing rabbits, why you have included Zn over this current recommendations.

Response 3: Thank you very much for your comments. The diet we used in the current experiment is the basal diet that we used for the rabbit’s farm in our university, and it included the same ingredients and premix as well. The analyzed zinc content in the basal diet added with ZnO in the premix was 90 mg/kg, which is still within the range of recommended zinc levels in rabbit diets. We have mentioned it in the introduction. Please see lines 61-63.   

Comments 4: In the introduction was mentioned that potential synergic effect with Zn sources, but it must be explained why alone acacia inclusion increased the Zn level in the animals.

Response 4: Thanks for your suggestion. According to your suggestions, we have included an explanation to the discussion area to further illustrate this point as much as possible using available study. Please see lines 342-347.     

Comments 5: Section 3.1. of the results is not a result section, I recommend including it at the material and methods section.

Response 5: Done as recommended. Moved as section 2.2. in the material and methods section. Please see lines 124-134.  

Comments 6: Were animals housed individually? The size of the cages seems to be too big for individual housing.

Response 6:  Thanks, it is recommended in our institution to provide more space for the animals. It is considered a good welfare program.

Comments 7: Please, indicate if the chemical composition was determined in your lab or provided by the feed manufacturer. Was DE value determined in a digestibility trial? Describe it.

Response 7: We appreciate your suggestions. The requested information is added. Please see lines 172-173 and 190 as a footnote in Table 3.

Comments 8: However, Zn level of the basal diet it is very important as the paper is focused in the Zn level and source. Please, describe how you have determined the Zn in the basal diet, as well as the Zn level of the four dietary treatments.

Response 8:  Thanks for your inputs. We have analyzed Zn levels in the basal diet, but we did not analyze it for the other treatments as Nano-ZnO was added on the top of the basal diet. Please see Table 3, and we explained this clearly for your earlier comment number 3. 

Comments 9: I was surprised by the Zn value of the basal diet (90 g/kg), I believe that it is a mistake as the current recommendations for growing rabbits are 35 mg/kg. Please, review this value and justify why the basal diet includes ZnO in the premix. Define that your paper is addressed to evaluate the inclusion of nano-ZnO providing Zn over the current recommendations.

Response 9: Thank you; it was written incorrectly and has been corrected in Table 3 (0.09 g/kg or 90 mg/kg). We have included an explanation in the introduction section about the recommended levels of Zn in rabbit diets based on recent studies to further illustrate this point. Please see lines 58-72.

Comments 10: Line 181 (n=20). Does it mean that you have slaughtered all the animals of each treatment??

Response 10: Yes, at the end of the study all animals were slaughtered. (n=20) added in line 214.

Comments 11: Justify in the introduction the need to control the effect of the treatments on both blood biochemical and carcass traits. 

Response 11: Thanks. We have, accordingly, revised introduction to include blood and carcass criteria. Please see lines 56-57, and 68-70.

Comments 12: Statistical analysis of performance traits (BW, ADG, DFI and FCR) must be done using a repeated measures procedure (mixed model), in order case, you are providing the effect of the animal within the treatment to treatment, overestimating the effect of the treatment.

Response 12: We used completely randomized design, and all the data was analyzed using GLM procedure of SAS using pens as the experimental unit. It is not wise to use repeated measure procedure for one set of data (in this case, performance data where we have only 3 repeated measurements), and then one-way ANOVA for other data. All data were evaluated for normal distribution (W > 0.05) using the Shapiro–Wilks test to ensure that there were normality distributions among the animals as a replicate. We have mentioned this in lines 228-229.  

Comments 13: All the tables and figures must be self-explanatory, please review the use of abbreviations in these elements.

Response 13: Thanks, done as suggested. Please see all Tables and Figures.

Comments 14: Avoid the use of the term “demonstrate” by “show”.

Response 14: Done as requested. See line 236.

Comments 15: In the discussion section there is a lack of comparison with the current recommendations of Zn and other previous studies where Zn level in rabbits has been evaluated. It is important due to the different level used in this study.

Response 15: Your suggestion is much appreciated. To further highlight this point in comparison to our study, we provided an explanation in the discussions section on the levels of Zn in rabbit diets based on recent studies. Please see the discussion section.   

Response to Comments on the Quality of English Language

Point 1: The English language is not so bad, but it could be reviewed by a native.

Response 1:  More proofreading was done for English language. Thank you very much for your positive opinion on our manuscript, and inputs which has improved the final form of our manuscript.

Reviewer 2 Report

In table 4, the average weight of rabbits at the age of 100 days for the subsequent experimental groups is: 2508, 2782, 2750, 2876, and in table 5 the average weight of rabbits at the age of 100 days is for the subsequent experimental groups: 2577, 2656, 2677, 2705, respectively. It should be explained what causes the differences in the final weight of rabbits in Tables 4 and 5. Only after clarifying this issue will it be possible to compare the impact of the preparations used on the average final weight of rabbits in the experimental groups and assess whether the differences are statistically significant.

Author Response

Thank you very much for taking the time to review this manuscript. Please find the detailed responses below and the corresponding revisions/corrections highlighted in track changes in the resubmitted files. We addressed all suggestions and comments.

Response to Reviewer 2 Comments

Thank you very much for taking the time to review this manuscript. Please find the detailed responses below and the corresponding revisions/corrections highlighted in track changes in the resubmitted files.

Comments 1: In table 4, the average weight of rabbits at the age of 100 days for the subsequent experimental groups is: 2508, 2782, 2750, 2876, and in table 5 the average weight of rabbits at the age of 100 days is for the subsequent experimental groups: 2577, 2656, 2677, 2705, respectively. It should be explained what causes the differences in the final weight of rabbits in Tables 4 and 5. Only after clarifying this issue will it be possible to compare the impact of the preparations used on the average final weight of rabbits in the experimental groups and assess whether the differences are statistically significant.

Response 1: Thank you for pointing this out. You are correct there are differences in the final weight of rabbits in Tables 4 and 5. The animals were weighed at 8:00 a.m. on day 100, and they were then fasted for 12 hours before slaughtering. This could explain the small differences in the rabbits' live body weight after 12 hours of fasting before they were slaughtered. This is mentioned in Lines 212-213. 

Reviewer 3 Report

The article “Effects of dietary Acacia nilotica fruit, zinc oxide nanoparticles and their combination on productive performance, Zn retention, and blood biochemistry of rabbits under hot climatic conditions” by Abdel-Wareth et al., described an interesting trial on rabbits.

I would like to suggest some minor revisions:

Title

Line 3: Zn, avoid to use the abbreviation

Line 4: under hot climate conditions .. how hot climate can inflence this trial ? there is not comparison with other climatic conditions… I would remove this words.

Simply Summar

Line 26: Zn, avoid abbreviation

Abstract

Line 29: Acacia nilotica, italicized

Line 32: What is the percentage of components and concentration of Nano-ZnO and ANFP mixture?

Line 35: Zn, avoid to use abbreviation

Introduction

Line 63: please rewrite the sentence, describing the size of this Zn Oxide particle.

Line 74: these activities were tested in vivo or in vitro? Please add some details.

Material and Methods

Line 105: Acacia nilotica, italicized

Line 139-142: please rephrase the sentence, avoiding mentioning the department twice.

Line 148: What is the percentage of components and concentration of Nano-ZnO and ANFP mixture? More details have to be added here.

Line 155: ad libitum, italicized

Line 184: please add more details about “absorption spectrophotometer”, at least the details of the instrument.

Author Response

Thank you very much for taking the time to review this manuscript. Please find the detailed responses below and the corresponding revisions/corrections highlighted in track changes in the resubmitted files. We addressed all suggestions and comments.

Response to Reviewer 3 Comments

Comments 1: The article “Effects of dietary Acacia nilotica fruit, zinc oxide nanoparticles and their combination on productive performance, Zn retention, and blood biochemistry of rabbits under hot climatic conditions” by Abdel-Wareth et al., described an interesting trial on rabbits.

Response 1: Thank you very much for taking the time to review this manuscript. Please find the detailed responses below and the corresponding revisions/corrections highlighted in track changes in the resubmitted files.

Comments 2: Title: Line 3: Zn, avoid to use the abbreviation

Response 2: Thank you for pointing this out. We agree with this comment. Changed as requested. Line 2 in Title.

Comments 3: Line 4: under hot climate conditions. how hot climate can influence this trial? there is not comparison with other climatic conditions… I would remove these words.

Response 3: Thanks, removed as suggested.

Comments 4: Line 26: Zn, avoid abbreviation

Response 4: Thanks, changed as requested, Line 25

Comments 5: Line 29: Acacia nilotica, italicized

Response 5: Thanks, done as suggested, Line 28

Comments 6: Line 32: What is the percentage of components and concentration of Nano-ZnO and ANFP mixture?

Response 6: added as requested, please see lines 31-32.

Comments 7: Line 35: Zn, avoid to use abbreviation.

Response 7: Thanks, changed as requested, line 34.

Comments 8: Line 63: please rewrite the sentence, describing the size of this Zn Oxide particle.

Response 8: Thanks, we have revised the sentences according to your suggestions in lines 58-70.

Comments 9: Line 74: these activities were tested in vivo or in vitro? Please add some details.

Response 9: Thank you for your comments. It was in vitro study. Added as suggested in line 79.

Comments 10: Line 105: Acacia nilotica, italicized

Response 10: italicized as requested, see line 109.

Comments 11: Line 139-142: please rephrase the sentence, avoiding mentioning the department twice.

Response 11: rephrased as requested, please see lines 157-160.

Comments 12: Line 148: What is the percentage of components and concentration of Nano-ZnO and ANFP mixture? More details have to be added here.

Response 12: Thanks, the requested information on the mixture is added in lines 164-167.

Comments 13: Line 155: ad libitum, italicized

Response 13: Thanks, italicized as requested, please see line 171.

Comments 14: Line 184: please add more details about “absorption spectrophotometer”, at least the details of the instrument.

Response 15: Agreed. We have added more details about “absorption spectrophotometer” as requested in Line 201. Thank you very much for your positive opinion on our manuscript and input which has improved the final form of our manuscript.

Reviewer 4 Report

Dear authors,

The manuscript was well-written and the content was informative and well-presented. I commend the authors for the comprehensive and systematic review of the topic. The manuscript will be a valuable contribution to this journal.

However, I’ve mentioned a few minor corrections that need to be corrected in the comment section of the main manuscript file. Some of these are the following:

Line 30: Please explain the reason behind choosing only male Californian rabbits, why not mixed-sex rabbits?

Line 31: Please also mention their average body weight before start of this experiment 

Please add one line at the end of the abstract, which basically explains the basic output of this study and the future recommendations related to this study work as well.

Line 58-59: Please explain a little bit more about what kind of grains, and how this phytate limit the absorption of Zn ?

Line 62: Please provide refernce to justify your statement 

Line 123-124: Please cite a reference here 

Line 183-184: Please provide the name of copmany, and country name for this spectrophotometer. 

Line 365:  Please explain a bit more in detail in this conclusion section, Future recommendations based on your current findings. 

Please check all the references' format should be according to the guidelines of the journal. 

Best wishes

Author Response

Thank you very much for taking the time to review this manuscript. Please find the detailed responses below and the corresponding revisions/corrections highlighted in track changes in the resubmitted files. We addressed all suggestions and comments.

Response to Reviewer 4 Comments

Comments 1: The manuscript was well-written and the content was informative and well-presented. I commend the authors for the comprehensive and systematic review of the topic. The manuscript will be a valuable contribution to this journal. However, I’ve mentioned a few minor corrections that need to be corrected in the comment section of the main manuscript file. Some of these are the following:

Response 1: Thank you very much for taking the time to review this manuscript. Please find the detailed responses below and the corresponding revisions/corrections highlighted in track changes in the resubmitted files.

Comments 2: Line 30: Please explain the reason behind choosing only male Californian rabbits, why not mixed-sex rabbits?

Response 2: Thank you for pointing this out. The main reason of using only male Californian rabbits for this study was to avoid the sexual productive and physiological variations (variations due to gender) among the results within the treatments. Moreover, our aims were to focus on the effects of the Nano-ZnO and acacia on the growth performance, liver and kidney functions, and zinc retention in tissues in male rabbits. We can take into consideration using an equal number of male and female rabbits in future studies, which is an excellent suggestion from your side.

Comments 3: Line 31: Please also mention their average body weight before start of this experiment 

Response 3: Thanks, added as suggested in abstract, and methods sections. Line 29-30 and 161-162, respectively.

Comments 4: Please add one line at the end of the abstract, which basically explains the basic output of this study and the future recommendations related to this study work as well.

Response 4: As suggested, we added at the end of abstract this point. Please see lines 41-43.

Comments 5: Line 58-59: Please explain a little bit more about what kind of grains, and how this phytate limit the absorption of Zn?

Response 5: Thanks. We have, accordingly, added requested information. Please see lines 58-61.

Comments 6: Line 62: Please provide reference to justify your statement.  

Response 6:  Done as requested, please see lines 61-70.

Comments 7: Line 123-124: Please cite a reference here 

Response 7: Done as requested, please see line 126.

Comments 8: Line 183-184: Please provide the name of company, and country name for this spectrophotometer. 

Response 8: Done as suggested in lines 202-203.

Comments 9: Line 365:  Please explain a bit more in detail in this conclusion section, Future recommendations based on your current findings. 

Response 9: Agreed. We have modified the conclusion to emphasize this point. Please see lines 387-391.

Comments 10: Please check all the references' format should be according to the guidelines of the journal. 

Response 10: We checked the references list and formatted according to the guidelines of the journal. Thank you so much for your comments and suggestions that helped to improve our manuscript.

Round 2

Reviewer 2 Report

Table 4 shows the weight of rabbits after finishing fattening at the age of 100 days, and Table 5 shows the weight of rabbits after 12 hours of fasting. Why is the weight of rabbits from the control group after fattening in Table 4 (2505g) lower than after the 12-hour fasting period in Table 5 (2577g)? In Table 4, the differences in the final weight of rabbits from different groups turned out to be statistically significant, while in Table 5 they are not statistically significant. These differences must be explained because they indicate the suitability of the preparation used for rabbit slaughter.

Author Response

Thank you very much for taking the time to review our manuscript again. Please find the attached detailed responses below and the corresponding revisions/corrections highlighted in track changes in the re-submitted files.
